# Circulating Serum Cystatin C as an Independent Risk Biomarker for Vascular Endothelial Dysfunction in Patients with COVID-19-Associated Multisystem Inflammatory Syndrome in Children (MIS-C): A Prospective Observational Study

**DOI:** 10.3390/biomedicines10112956

**Published:** 2022-11-17

**Authors:** Marcela Kreslová, Petr Jehlička, Aneta Sýkorová, Daniel Rajdl, Eva Klásková, Pavel Prokop, Sabina Kaprálová, Jan Pavlíček, Romana Kaslová, Alžběta Palátová, Veronika Mohylová, Josef Sýkora

**Affiliations:** 1Department of Pediatrics, Faculty Hospital, Faculty of Medicine in Pilsen, Charles University in Prague, Alej Svobody 80, 30460 Pilsen, Czech Republic; 2Department of Clinical Biochemistry and Hematology, Faculty Hospital, Faculty of Medicine in Pilsen, Charles University in Prague, Alej Svobody 80, 30460 Pilsen, Czech Republic; 3Department of Pediatrics, Faculty Hospital, Faculty of Medicine and Dentistry, Palacky University in Olomouc, I. P. Pavlova 185/6, 77520 Olomouc, Czech Republic; 4Department of Pediatrics, Faculty Hospital, Faculty of Medicine, Ostrava University, 17. Listopadu 1790, 70852 Ostrava, Czech Republic

**Keywords:** endothelial dysfunction, multisystem inflammatory syndrome, reactive hyperemia index, cystatin C, atherosclerosis, coronavirus disease 2019, biomarkers, asymmetric dimethylarginine, severe acute respiratory syndrome coronavirus 2

## Abstract

Introduction: Multisystem inflammatory syndrome in children (MIS-C) is a new clinical entity that has emerged in the context of the COVID-19 pandemic. Despite the less severe course of the disease, varying degrees of cardiovascular events may occur in MIS-C; however, data on vascular changes occurring in MIS-C are still lacking. Endothelial dysfunction (ED) is thought to be one of the key risk factors contributing to MIS-C. Background: We conducted a prospective observational study. We investigated possible manifestations of cardiac and endothelial involvement in MIS-C after the treatment of the acute stage and potential predictive biomarkers in patients with MIS-C. Methods: Twenty-seven consecutive pediatric subjects (≥9 years), at least three months post-treated MIS-C of varying severity, in a stable condition, and twenty-three age- and sex-matched healthy individuals (HI), were enrolled. A combined non-invasive diagnostic approach was used to assess endothelial function as well as markers of organ damage using cardiac examination and measurement of the reactive hyperemia index (RHI), by recording the post- to pre-occlusion pulsatile volume changes and biomarkers related to ED and cardiac disease. Results: MIS-C patients exhibited a significantly lower RHI (indicative of more severe ED) than those in HI (1.32 vs. 1.80; *p* = 0.001). The cutoff of RHI ≤ 1.4 was independently associated with a higher cardiovascular risk. Age and biomarkers significantly correlated with RHI, while serum cystatin C (Cys C) levels were independently associated with a diminished RHI, suggesting Cys C as a surrogate marker of ED in MIS-C. Conclusions: Patients after MIS-C display evidence of ED, as shown by a diminished RHI and altered endothelial biomarkers. Cys C was identified as an independent indicator for the development of cardiovascular disease. The combination of these factors has the potential to better predict the cardiovascular consequences of MIS-C. Our study suggests that ED may be implicated in the pathophysiology of this disease.

## 1. Introduction

Severe acute respiratory syndrome coronavirus 2 (SARS-CoV-2) infection is a systemic inflammatory pandemic disease predominantly affecting the respiratory system. It was originally thought to cause a viral pneumonia leading to acute respiratory failure. However, the evolving clinical and laboratory findings suggest a crucial role of altered endothelial function in its pathophysiology, contributing to multiorgan dysfunction [1]. SARS-CoV-2 infection in children has an overall better prognosis than in adults [2].

Multisystem inflammatory syndrome in children (MIS-C) is a novel, serious phenomenon that has emerged in the context of the SARS-CoV-2 pandemic in the pediatric population, first reported in April 2020 [3]. MIS-C can appear 2–6 weeks after SARS-CoV-2 infection. Although the pathophysiology of MIS-C has not been fully understood yet, it is assumed to result from an inadequate late immune response to the virus, with some clinical similarities to Kawasaki disease. Common presenting symptoms include persistent fever, shock state, and gastrointestinal, mucocutaneous, renal, and cardiorespiratory involvement. In 50% of documented MIS-C cases, children had myocardial impairment presenting with depressed left ventricular function, coronary artery dilation or aneurysms, myocarditis, pericarditis, refractory shock with troponin elevation, or N-terminal fragment of brain-type natriuretic peptide (NT-proBNP). Treatment strategies include intensive care admission with fluid resuscitation, vasoactive circulatory support, administration of intravenous immunoglobulins (IVIG) and corticosteroids (CS), antiplatelet, and in some cases, anticoagulant therapy. In the case of early and intensive treatment, the outcome is generally very good. Death due to MIS-C is rare, with a mortality rate of 1.4–1.9% [4]. The severity of the clinical condition is mainly related to myocardial involvement and circulatory instability (hypotension, shock) [5]. However, it is currently not entirely clear what possible cardiovascular consequences this disease may have in surviving patients over a longer period of time.

Endothelial dysfunction (ED) is a systemic disorder and it might be considered as an integrated marker of all atherogenic and atheroprotective factors present in an individual, variables, and genetic predispositions [6,7]. ED represents a key variable in the development of atherosclerosis and its complications. The monolayered endothelium is the main regulator of blood clotting, vascular tone, as well as immune and inflammatory processes by generating and balancing various active molecules. ED is a condition in which the endothelial layer of small arteries fails to effectively perform its normal function [1]. ED is also characterized by reduced vasodilative responses to an appropriate ischemic stimulus [8]. In addition, serum cystatin C (Cys C) has been proposed as a potential biomarker of increased cardiovascular (CV) risk [9]. Infections are one of the several varied triggers of ED. The current studies [1,10,11] on ED in SARS-CoV-2 infection highlight its pathophysiology through the direct viral effect, which induces endothelial injury, through cytokine response, oxidative stress, uncontrolled immune and inflammatory response, coagulation disturbance, and their interactions, resulting in a vicious cycle aggravating the disease process. The reversibility of ED is a primary target in the effort to optimize therapeutic strategies and reduce CV risk in children. Early detection of this process might have therapeutic and prognostic implications [12].

The nature of the SARS-CoV-2 infection and MIS-C requires very careful monitoring. One of the areas is the assessment of the CV system. There is a lack of data regarding the CV risk assessment and the identification of possible biomarkers to predict the severity of complications in children after MIS-C [13].

### Aims

To fill the knowledge gap on CV risk and address limitations of existing studies, we conducted a non-interventional observational study focusing on CV involvement in children (≥9 years) after the treatment of the acute-stage MIS-C. Further, we added enhanced insights into biomarkers of ED that could help estimate the risk of long-term complications of the disease and provide an opportunity for better patient management and the possibility to focus on prevention.

The aims of the study were: (1) to compare the reactive hyperemia index (RHI) in patients after MIS-C and healthy individuals (HI), (2) to determine the relationship between a non-invasive combined diagnostic approach and risk factors for cardiovascular disease (CVD), such as MIS-C severity characterized by the length of intensive care unit (ICU) hospitalization and the length of administration of catecholamines, as well as the time interval between MIS-C and sampling, and lung function, and (3) to find possible biomarkers for predicting the increased CV risk in patients with MIS-C.

## 2. Materials

### 2.1. Study Population

This prospective, observational, multicenter cohort study was conducted during the 2021–2022 season at three Departments of Pediatrics in the Czech Republic: Faculty Hospital, Faculty of Medicine in Pilsen, Charles University in Prague; Faculty Hospital, Faculty of Medicine and Dentistry, Palacky University in Olomouc; Faculty Hospital, Faculty of Medicine, Ostrava University in Ostrava.

A total of 27 pediatric Caucasian subjects older than 9 years, at least 3 months after MIS-C treatment of varying severity, in stable condition, without clinical signs of MIS-C, with no signs of acute infection, and no known vascular disease, were enrolled during the 1-year study. All subjects participated in the study voluntarily and provided written informed consent. The minimum interval from MIS-C was set at three months to assess late CV complications. The diagnosis of MIS-C was made according to the guidelines for this clinical unit after excluding other more common conditions that cause similar symptoms, such as Kawasaki disease, bacterial sepsis, and toxic shock syndrome, and after SARS-CoV-2 infection was confirmed by a positive reverse transcription polymerase chain reaction, serology, antigen test, and/or exposure to SARS-CoV-2 infection within 4 weeks before the onset of MIS-C symptoms [14]. All MIS-C subjects were treated according to current MIS-C standards [15].

The comparison group consisted of 23 age- and sex-matched healthy subjects with no history of chronic disease known to affect microvascular function, inflammatory, neoplastic, metabolic, cardiac, or peripheral vascular disease, as well as no anti-inflammatory, antibiotic, and vasoactive treatment affecting endothelial function. HI recruitment was performed in the same period and all subjects were examined uniformly to avoid problems of bias arising from inappropriate controls. With the help of an experienced cardiologist, a detailed questionnaire system was developed and used to collect the necessary data on familial CVD risk.

Exclusion criteria for both groups were a positive family history of premature CV events, abnormal left ventricular function, dyslipidemia, obesity (body mass index (BMI) > 30), age under 9 years, chronic medication, smoking, and substance abuse. A positive family history of premature manifestation of CVD was defined as the occurrence of sudden death or myocardial infarction in the father or a first-degree male relative before the age of 45, and the occurrence of sudden death or myocardial infarction in the mother or a first-degree female relative before the age of 55. The exclusion of smokers and obese subjects from the comparison cohort is consistent with the demonstration of increased baseline pulse amplitude in obese people with metabolic syndrome and an inverse relationship between baseline amplitude and peripheral arterial tone (PAT) response to hyperemia [8].

### 2.2. Methods

A combined non-invasive diagnostic approach was used to assess endothelial function and related cardiac complications to elucidate CV risk in children and adolescents after MIS-C. Cardiac evaluation, RHI measurement, and biochemical methods in relation to ED and heart disease were used to detect early changes in endothelial function in children treated for MIS-C. For all MIS-C patients, data collection included the duration from the first symptoms of the MIS-C disease as well as the length of ICU hospitalization, the need for ventilation support, the period of circulatory support, pharmacologic treatment such as CS, IVIG, catecholamines, antibiotics, antiplatelet drugs, anticoagulants, and adverse CV outcomes, if available.

At admission, weight, height, and blood pressure were measured. Nutritional status was assessed using BMI. The physical and mental condition after MIS-C was evaluated by a questionnaire system. All participants completed a questionnaire assessing physical performance with an emphasis on any post-exercise difficulties, mental health, sleep, fine motor skills, smell, taste, hearing, and joints, with the possible addition of other post-COVID problems of each individual. Symptoms before and 3 months after MIS-C were compared. Endothelial function was assessed by non-invasive plethysmographic examination determining the EndoPAT index (RHI), by recording the post- to pre-occlusion pulsatile volume changes and evaluating biomarkers in relation to ED: high-sensitive C-reactive protein (hsCRP), asymmetric dimethyl arginine (ADMA), symmetric dimethyl arginine (SDMA), and ADMA/SDMA. Furthermore, a list of other biomarkers putatively influencing ED was assessed: kidney function and injury (plasmatic creatinine and Cys C, albuminuria), hepatic function (bilirubin, alanine transaminase (ALT), alkaline phosphatase (ALP), lactate dehydrogenase (LD)), inflammation (complete blood count (CBC), procalcitonin, interleukin-6 (IL-6), ferritin), coagulation (D-dimer, fibrinogen), and fibrotization and cellular stress (soluble suppression of tumorigenicity 2 (sST2), growth differentiation factor-15 (GDF-15)). The cardiac examination included an assessment of the functional and structural state of the heart using echocardiography, electrocardiogram (ECG), and examination of laboratory parameters, especially high-sensitive troponin T (hsTNT) and NT-proBNP. In addition, lung function was evaluated by spirometric forced expiratory volume in one second (FEV_1_) using the Flowhandy ZAN 100 (nSpire Health NT-proBNP, Inc., Oberthulba, Germany).

### 2.3. Cardiac Evaluation

In all participants, a standardized transthoracic echocardiographic study was completed. Standard parasternal, apical, and subcostal views were obtained in accordance with the American Society of Echocardiography guidelines [16] using the EPIQ 7 device (Philips Healthcare, Amsterdam, Netherlands) or the Vivid^TM^ E95 device (GE Healthcare, Wauwatosa, WI, USA). Echocardiographic examination comprised the evaluation of intracardiac hemodynamics as well as heart anatomy, left and right ventricle functions, and assessment of heart valve function. Systolic function of both ventricles was determined in M-mode, based on the shortening fraction (SF) obtained from parasternal long-axis in the left ventricle, and tricuspid annular plane systolic excursion (TAPSE) in the right ventricle. Decreased left ventricular systolic function was defined as FS < 25%, and a value of TAPSE < 10 mm was indicative of RV systolic dysfunction. The diameters of the proximal segments of the right and left coronary arteries were assessed in the standard parasternal short-axis view. Size of coronary arteries was classified according to internal lumen diameter, and all measurements were normalized to body surface area, and the Z-score of all diameters was calculated according to the current echocardiographic nomograms for the pediatric population. Dilation was defined as Z-score 2 to <2.5, small aneurysm as Z-score ≥ 2.5 to <5, medium aneurysm as Z-score ≥ 5 to <10 and absolute dimension < 8 mm, and giant aneurysm as Z-score ≥ 10 and absolute dimension ≥ 8 mm [17].

The cardiac examination was complemented by standard 12-lead electrocardiography (ECG) using the BTL-08 SD3 (BTL Industries Ltd., Hertfordshire, United Kingdom) and MAC^®^ 1600 ECG Analysis System devices (GE Healthcare, Wauwatosa, WI, USA), as well as blood pressure and laboratory parameters, especially hsTNT and NT-proBNP.

### 2.4. Reactive Hyperemia Index Measurements

The methodology for measuring digital vascular function using PAT was described in detail in our previous article [18]. RHI was assessed using the Endo-PAT^TM^2000 device (Itamar Medical Ltd., Caesarea, Israel) based on non-invasive plethysmographic measurements of arterial tone changes in peripheral arterial vessels. PAT technology is based on recording the endothelium-mediated changes in vascular tone (post- to pre-occlusion pulsatile volume changes). The biosensor is placed on the index fingers of each hand and non-invasively measures finger arterial pulsatile volume changes by collecting the PAT signal. The measurement takes place with the examinees lying on their backs in a quiet, thermoneutral room. First, weight, height, and blood pressure are measured in the non-dominant arm. During the first phase, PAT input values are recorded on the upper limbs. During the following five-minute second phase of the examination, the pressure cuff is inflated 60 mm Hg above the examinee’s systolic pressure, to a minimum of 200 mm Hg, which results in occlusion of the brachial artery on the non-dominant limb and causes a subsequent hyperemic response. During the third phase, subsequent deflation of the cuff results in a sudden flood of blood, causing endothelium-dependent flow-mediated dilatation, and the post-occlusive endothelium-induced dilation with reactive hyperemia is captured as an increase in the PAT signal amplitude. The EndoPAT index (RHI) is automatically calculated from the ratio of occlusive and pre-occlusive arterial flow relative to the values of simultaneously measured, non-occluded contralateral limbs. An insufficient increase in PAT amplitude during the last post-occlusive phase is associated with ED. The RHI threshold for adults is set at 1.67. A value <1.67 is considered pathological, while a score above 2.10 is recommended [19]. The measurement is also performed from the contralateral upper limb to control simultaneous changes in vascular tone independent of the endothelium.

### 2.5. Laboratory Methods

Laboratory analysis included examination of venous blood (12 mL in total) and urine. Three tubes with anticoagulant additives were obtained: K_3_EDTA for CBC, sodium citrate for coagulation D-dimers and fibrinogen analysis, and lithium heparin for all other analyses. Urine was collected into a tube without any additives, and urinary albumin and creatinine were determined from this material. CBC determination (performed in the laboratories of the participating centers) and the processing of the collected tubes were carried out within one hour after the collection. Sodium citrate tubes were centrifuged for 10 min at 1500× *g* and the supernatant without a buffy coat was centrifuged at the same conditions to obtain platelet-poor plasma, and then the supernatant was transferred to a new tube. Lithium heparin tubes were centrifuged for 10 min at 1500× *g* and the supernatant was transferred to new tubes, and all other parameters were determined from this material. Samples were frozen at –80 °C until analysis was performed (after thawing) batch-wise, in the central laboratory at the Department of Clinical Biochemistry and Hematology, Faculty Hospital in Pilsen.

#### 2.5.1. ADMA and SDMA Quantification

The analysis was performed on the Dionex UltiMate 3000 UHPLC-Standard (Thermo Fisher Scientific, Waltham, MA, USA) with ion trap AmazonSL (Bruker, Billerica, MA, USA) and nitrogen generator GENIUS NM32LA (Peak Scientific Instruments, Inchinnan, Great Britain). Separation was performed on the Kinetex 2.6 µm C18 100A HPLC Column 50 × 2.1 mm with the precolumn Kinetex UHPLC C18 2.1 mm. Mobile phase A consisted of methanol and 0.1% acetic acid and mobile phase B consisted of 5 mmol/L of ammonium acetate and 0.1% acetic acid (pH = 4.3). Gradient elution was used with a total run time of 8 min. The initial composition of mobile phases was 10% A, with a linear increase to 100% A in 3 min. Composition of 10% A returned within 4 min. The flow rate was 0.25 mL/min with a 20 µL injection volume. The temperatures of the autosampler and column thermostat were 6 and 20 °C. Every measurement of sample/calibrator/quality control was performed in duplicate.

Mass spectrometry ionization and fragmentation of dimethyl derivatives and internal standards were optimized on an ion trap mass spectrometer with electrospray ionization in a positive mode. The ion source temperature and desolvation temperature were 180 °C. Nitrogen was used as a nebulizer (4.0 L/min, 7.3 psi) and desolvation gas. Helium (purity 5.0) was used as collision gas and set at 3.5010–6 mbar. The capillary voltage was 5000 V. Quantification was performed in the multiple reaction monitoring mode (scan mode: enhanced resolution).

The method was created in the software HyStar (ver. 3.2, © Bruker Daltonik GmbH). The chromatographic part of the method was controlled by Chromeleon SW (ver. 6.8, SR12, © 1994–2013 Dionex Corporation, Part of Thermo Fisher Scientific) and the ion trap was controlled by TrapControl SW (ver. 7.2, © 1998–2013 Bruker Daltonik GmBH). The software products for the evaluation of measured data were DataAnalysis (ver. 4.2, © 1993–2013 Bruker Daltonik GmBH) and QuantAnalysis (ver. 2.2, © 1999–2013 Bruker Daltonik GmBH). The coefficients of variation (CoV) in our ADMA and SDMA measurements were 5.63% and 4.9%.

#### 2.5.2. Special Parameters’ Determination

Levels of sST2 were measured using a ST2/IL-33R Quantikine ELISA Kit (R&D Systems, Minneapolis, MN, USA) on a ThunderBolt ELISA Analyzer (Gold Standard Diagnostics, Davis, CA, USA). CoV for sST2 was 6.27%.

GDF-15 (CoV 1.59%), IL-6 (CoV 2.40%), ferritin (CoV 2.90%), hsTnT (CoV 2.56%), and procalcitonin (CoV 2.06%) were determined by the ECLIA Immunoassay (Roche, Mannheim, Germany) on a cobas e602 analyzer (Roche, Mannheim, Germany).

#### 2.5.3. Routine Parameters’ Determination

Sodium, potassium, and chlorides in plasma and urine were determined by ISE modules of a Roche c702 analyzer. Bilirubin, ALT, ALP, albumin, creatinine, creatine kinase (CK), and LD were determined by routine photometric kits from Roche (Roche, Mannheim, Germany) on a cobas c702 analyzer (Roche, Mannheim, Germany). HsCRP (CoV 3.12%) and Cys C (CoV 4.15%) were determined by the immunoturbidimetric principle on a cobas c702 analyzer (Roche, Mannheim, Germany), and urinary albumin on a cobas c501 analyzer (Roche, Mannheim, Germany).

CBC was determined by blood particle analyzers DxH 900 (Beckman Coulter Inc., Brea, CA, USA) and XN-1000 (Sysmex Corporation, Kobe, Japan).

Quantitative determination of D-dimers in citrate plasma was assessed by the immunoturbidimetric method with HemosIL D-Dimer HS 500 and quantitative determination of fibrinogen in citrate plasma by the coagulation method according to Clauss with the HemosIL Q.F.A. thrombin kit with coagulometer ACL TOP 750 CTS (both kits and analyzers are from Instrumentation Laboratory, Bedford, MA, USA).

### 2.6. Statistical Analysis

Statistical data analysis was performed using SAS software (SAS Institute Inc., Cary, NC, USA) and all graphs were made using STATISTICA software (StatSoft, Inc., Tulsa, OK, USA).

Basic statistical data such as mean, standard deviation, variance, median, interquartile range, minimum, and maximum were calculated for the measured parameters. For categorical variables, their absolute and relative frequencies were calculated.

The matching of age, sex, systolic and diastolic blood pressure, and BMI between the patient and HI groups was tested using tests of equivalence (TOST) to verify the similarity of the two groups. The optimal RHI cutoff value for the pediatric population was calculated based on the specificity and sensitivity of the RHI factor between HI and MIS-C patients.

Relationships between variables were examined using correlation coefficients (Pearson) and linear regression. In cases where Pearson’s correlation coefficient showed a strong relationship between the investigated parameters and non-parametric ANOVA showed a statistically significant difference in the distribution of the investigated parameters, an appropriate cutoff was sought.

The clinical impact of individual factors on treatment success was expressed using specificity, sensitivity, positive predictive value, negative predictive value, and odds ratio. Differences in frequencies were tested using Fisher’s exact test or the Chi-square test. Multivariate analysis was performed using multiple regression and logistic regression (stepwise regression).

For all analyses, a *p*-value ≤ 0.05 was considered statistically significant.

## 3. Results

### 3.1. Demographic Data of the Study Population

A total of 27 pediatric Caucasian subjects over 9 years of age (median age 12.76 years at the time of examination), in a stable condition, with a good nutritional status (median BMI 20.82 kg/m^2^), at least 3 months after treatment for MIS-C of varying severity, were examined. Males accounted for 55.6% of patients. HI characteristics are shown in Table 1. HI matched the MIS-C group in terms of number and representation of sex, age, and ethnicity. Blood pressure values are shown in Table 1 and correlation analyses with RHI and Cys C are shown in Table 2. Systolic blood pressure values were normal in both groups (median 115 vs. 116; *p* = 0.702). In individuals with MIS-C, diastolic blood pressure values were statistically significantly higher (median 70 vs. 66; *p* = 0.009), but within the normal range for the subjects’ age (median 70 mmHg).

Group characteristics according to selected criteria (duration from the first symptoms of MIS-C, duration of ICU hospitalization, and administration of catecholamines, antiplatelet, and anticoagulant therapy, lung function assessed by spirometric FEV_1_) are shown in Table 2. Concerning duration (time between examination and first symptoms of MIS-C), the median was 11.85 months, with a minimum of 3.62 months. One patient experienced two reinfections with SARS-CoV-2 between MIS-C diagnosis and study examination.

Due to the severity of the clinical course of MIS-C, 22 patients (81.5%) required hospitalization in the ICU, with a median length of stay of 2 days and a maximum of 33 days. One patient (3.7%) required mechanical ventilation. Fourteen patients (51.9%) did not need catecholamine treatment. On the contrary, nine patients (33.3%) were treated with dual-combination therapy (25.9% dobutamine and noradrenaline, 7.4% dobutamine and dopamine) and four patients (14.8%) were given monotherapy (11.1% dobutamine, 3.7% noradrenaline). The maximum duration of catecholamines administration was 105 h (dobutamine). The longest ICU hospitalization was in a girl aged 14.3 years, with the first-degree heart block on ECG, initial hsTnT levels 268 ng/L, maximum hsTnT levels 449 ng/L, initial and peak NT-proBNP levels 10,702 ng/L, and SF 45% on echocardiography. This patient was treated with mechanical ventilation for five days, catecholamines (dobutamine 80 h, noradrenaline 100 h), CS, and IVIG, antiplatelet therapy for 19 days, and anticoagulants for 23 days. She complained of nail peeling after MIS-C. The RHI examination could not be completed due to her discomfort from the pressure of the inflated cuff on the arm.

Regarding antiplatelet therapy, 2 patients (7.4%) were not treated with it, while 25 patients (92.6%) were treated with antiplatelet agents (acetylsalicylic acid) with a median duration of therapy of 37 days. Regarding anticoagulants, another 2 patients (7.4%) were not treated with it, while 25 patients (92.6%) were treated with low molecular weight heparins (44.4% enoxaparin, 48.2% nadroparin) with a median duration of administration of 8 days. Regarding pharmacotherapy, only one patient was not given antibiotic treatment, and 96.3% of patients were treated with antibiotics (mainly cephalosporins). Twenty-five patients (92.6%) were treated with CS and the same number received IVIG. Regarding spirometry, all patients at least three months after MIS-C had a normal FEV_1_ as a standard parameter of lung function in clinical trials.

A characteristic comparison of clinical symptoms before and three months after MIS-C treatment is shown in Table 3. Transient deterioration of physical performance was found in 37% of patients, but the condition normalized in all of them and showed good tolerance of physical activity at the time of examination. Post-exercise difficulties (shortness of breath) were reported in one patient (3.7%). Two patients suffered from sleep disorders (7.4%). Other symptoms such as heart palpitations, swelling of the ankles, and nail peeling were reported by 11.1% of examined patients. No one reported mental disorders, deterioration of fine motor skills, hearing disorders, or joint pain.

Furthermore, red blood cell transfusion was required in two cases due to anemia (hemoglobin levels were 70–75 g/L). Acute renal failure occurred in three patients.

### 3.2. Cardiac Outcomes

Evaluation of initial cardiac parameters such as hsTnT and NT-proBNP levels in patients with MIS-C as well as positivity of the echocardiographic findings are shown in Table 4. Median initial hsTnT levels were 45 ng/L, and the median NT-proBNP was 2612 ng/L. During MIS-C treatment, minimum and maximum hsTnT levels in our patients were the same, with a median of 45 ng/L. Median peak NT-proBNP levels during hospitalization were 4269 ng/L (min 107, max 35,000).

Sixty-three percent of the total MIS-C patients (17 patients) had cardiac involvement, detailed in Table 5. In our group, left ventricular dysfunction (LVD) was confirmed in 10 patients (37%). Signs of heart failure were demonstrated in two patients (7.4%). Four patients (14.8%) met the echo criteria of mitral and aortic valvulitis. Coronary artery dilatation was detected in four patients (14.8%). Arrhythmias were confirmed in three patients (11.1%). All patients recovered and are free of complications in the medium term.

In our MIS-C cohort, in terms of coronary artery evaluation, none had aneurysm or coronary artery ectasia. Neither pulmonary embolism nor pulmonary hypertension was detected in any patient. Twenty-five patients (81.5%) had normal electrocardiogram results, and in one case first-degree heart block was found. Furthermore, second-degree heart block of the Wenckebach type was confirmed in an 18-year-old boy. In this patient, the initial rhythm on admission was significant first-degree heart block with a PR interval of about 220 milliseconds, intermittently interspersed with second-degree Wenckebach-type heart block. The average heart rate was 99 beats per minute. Since the patient was an active athlete, regular checks including ECG were performed and no recurrence of arrhythmia was detected.

Among the thrombotic complications, intracardiac thrombosis (thrombosis at the tip of the left ventricle with multiple peripheral embolizations, including the brain) was demonstrated in one 18-year-old boy. At admission, there were already signs of acute cardiorenal failure, circulatory instability, and echocardiographically without dilation of the coronary arteries, hsTnT 190 ng/L, NT-proBNP 35,000 ng/L, and D-dimer 11.0 mg/L FEU. On the second day of hospitalization, a short course of ventricular tachycardia (eight complexes) was detected without endangering the blood circulation without recurrence, and further with evidence of intracardiac thrombosis.

Regarding RHI, the plethysmographic examination could not be successfully completed in 3 patients out of a total of 27 examined, due to technical reasons or negative subjective feelings of pediatric patients during inflation of the pressure cuff on the arm. Therefore, statistical data processing was based on the results of 24 patients with MIS-C.

### 3.3. RHI, Laboratory Parameters, and Biomarkers

RHI values were statistically significantly lower in MIS-C (1.32 vs. 1.80; *p* = 0.001), as shown in Table 1 and Figure 1. Given the evidence of a statistically significant difference in RHI between groups, we sought a cutoff in our MIS-C cohort. With the receiver operator characteristic (ROC) area under the curve of 80% (Figure 2), we sought the statistically strongest clinical cutoff between the MIS-C and HI groups and found it at the RHI value of 1.40, with a specificity of 86.96%, a sensitivity of 66.67%, an odds ratio of 13.33, and 95% confidence limits (CL) of 3.03–58.52. Using the Chi-square test, we proved the frequency difference between groups with RHI below and above 1.40 (*p* = 0.002). Our results indicate that children and adolescents (≥9 years) in our cohort with RHI ≤ 1.40 have a 13.33 times higher ED risk.

Representation of cardiac, laboratory parameters, and biomarkers in MIS-C patients with correlation with RHI and Cys C is shown in detail in Table 6.

All examined MIS-C patients had IL-6 values under the limit of quantification of the method, i.e., <1.5 ng/L. Neutrophil and leucocyte counts were within age-matched reference ranges in all studied individuals. The neutrophil–leucocyte ratio median (min, max) value was 1.36 (0.78, 2.88)—far from the proposed cutoff value of 5.03 for MIS-C prediction [20]. All examined MIS-C patients had IL-6 values under the limit of quantification of the method, i.e., <1.5 ng/L.

### 3.4. Correlation Analysis

In a group of children and adolescents after MIS-C treatment, we demonstrated correlations of RHI with four markers: age, ADMA, ALP, and Cys C. We did not confirm correlations of RHI with nutritional status (BMI), lung function (FEV_1_), and clinical parameters (duration, blood pressure, ICU days, duration of catecholamines and anticoagulants).

#### 3.4.1. RHI Correlations with Age, ADMA, and ALP

Statistically significant correlations of RHI with other parameters were demonstrated: a positive moderate correlation of RHI with age (r = 0.508; *p* = 0.011), a negative low correlation of RHI with ADMA (r = –0.428; *p* = 0.037), and a negative moderate correlation of RHI with ALP (r = –0.555; *p* = 0.005). These correlations were without a statistically significant difference between groups, with RHI ≤ 1.4 and RHI > 1.4; therefore, we do not consider them clinically significant.

#### 3.4.2. RHI Correlation with Cys C

We found a negative moderate, statistically significant correlation of RHI with Cys C (r = –0.540; *p* = 0.006), as shown in Table 6 and Figure 3. We confirmed the difference in groups with RHI ≤ 1.4 and RHI > 1.4 using the Wilcoxon two-sample test (*p* = 0.045), as shown in Figure 4, and looked for a clinical cutoff between the groups, with the ROC area under the curve of 77% (Figure 5). We found the statistically strongest cutoff at a Cys C value of 0.89 mg/L, with a specificity of 75%, a sensitivity of 75%, an odds ratio of 9, and 95% CL of 1.62–21.54. Using Fisher’s exact test, we demonstrated a frequency difference between groups with Cys C below and above the value of 0.89 mg/L (*p* = 0.032). Statistically proven results indicate that in our cohort of children and adolescents (≥9 years), patients with Cys C above 0.89 mg/L have a 9 times higher risk of ED.

Finally, multivariate data analysis was performed using multivariate and logistic regression (stepwise regression). We confirmed a statistically significant Cys C factor, where higher Cys C values mean lower RHI values (*p* = 0.041).

#### 3.4.3. Cys C Correlations with Other Parameters

We tested Cys C for correlation with the following factors: RHI, biomarkers (hsCRP, ADMA, SDMA, ADMA/SDMA), cardiac parameters (LVDd, TAPSE, SF, hsTnT, NT-proBNP), clinical parameters (duration, age, BMI, blood pressure, FEV_1_, ICU days, duration of catecholamine therapy, anticoagulants, and antiplatelet agents), and laboratory parameters (ferritin, D-dimer, fibrinogen, GDF-15, IL-6, procalcitonin, sST2, Na, K, CI, S-creatinine, bilirubin, ALT, ALP, S-albumin, CK, LD, U-Na, U-K, U-CI, U-creatinine, U-albumin, urine albumin-creatinine ratio, leukocytes, lymphocytes, neutrophils).

Statistically significant correlations of Cys C with the following parameters were confirmed: a negative moderate correlation with RHI, also mentioned in Section 3.4.2 (r = −0.540; *p* = 0.006), a positive moderate correlation with antiplatelet agents (r = 0.566; *p* = 0.004), a negative moderate correlation with ALT (r = −0.513; *p* = 0.010), as shown in Figure 6: patients with higher ALT tend to have lower Cys C values, a positive weak correlation with SDMA (r = 0.458; *p* = 0.024): patients with higher SDMA tend to have higher Cys C values, and a positive weak correlation with creatinine (r = 0.411; *p* = 0.046): patients with higher serum creatinine levels tend to have higher Cys C values.

## 4. Discussion

Currently, there is no publication reporting a reasonable estimate of ED and characteristics of children with MIS-C available. In this study, a significantly higher microvascular ED in the MIS-C study cohort compared with HI has been shown for the first time. The cutoff point of RHI ≤ 1.4 was independently associated with a higher risk of CVD. Significant correlations between age, biochemical parameters, and RHI values were found in the present study. Even more, higher levels of Cys C were strongly and independently associated with an increased ED risk, suggesting Cys C as a surrogate marker of ED in patients with MIS-C. Thus, the addition of ED assessed by RHI can contribute to CVD risk prediction and stratification in patients after MIS-C. However, the determinants of prognosis in patients with MIS-C remain largely unexplored. It is worth noting that post-MIS-C patients still exhibit ED. Although the pathogenesis of CVD in MIS-C patients is still poorly understood, this suggests that severe impairment of endothelial function in peripheral microvasculature may be a key component of MIS-C. Our data may provide new options for research on pathological processes responsible for advanced ED. As such, therapeutics targeted at ED and its sequelae in treating MIS-C and its long-lasting effects are likely to be effective in preventing CVD.

Cys C is a biomarker belonging to a group of cysteine protease inhibitors produced primarily by nucleated cells [21]. Several sets of data from the literature demonstrate that levels of Cys C provide a valuable alternative marker to creatinine-based criteria for measuring renal function, particularly in the detection of small reductions in the glomerular filtration rate [22,23]. Additionally, the potential of Cys C as a novel predictive marker of CVD was deduced from a variety of studies [9,24,25,26].

To the best of our knowledge, no pediatric study has yet addressed the relationship between ED and Cys C levels in MIS-C patients. We showed that Cys C levels strongly correlated with ED among patients with MIS-C. Of interest in this study is the demonstration that after adjusting for confounding factors, we have found that higher Cys C levels were independent predictors of ED in patients with MIS-C. Furthermore, these data provide support for the link and further studies of Cys C as a surrogate marker of ED in patients with MIS-C, probably beyond renal function [9]. As far as we are concerned, the underlying mechanism of the link between high Cys C levels and CVD is largely unknown and hypothetical. According to Demirkol et al., Cys C levels should correlate with ED and inflammation indirectly through renal function [27]. A paper by Gelzo and co-workers adds potentially valuable information to the discussion [28]. The study group confirmed epithelial involvement in acute-phase MIS-C. We suggest that it could be potentially plausible that the unifying concept between Cys C as a potential inflammatory marker [29] and CVD might be micro-inflammation [9,30,31]; however, the role of Cys C and the possible connection of inflammatory pathways need to be further clarified before a definitive conclusion is made.

We hypothesize that combining the above-mentioned parameters significantly increases the importance of Cys C usage in clinical practice, may have predictive reliability, may be useful to identify patients with ED, and may improve risk stratification in MIS-C. No previous report has discussed the association between peripheral ED and Cys C concentration in MIS-C. A RHI ≤ 1.4, representing Cys C concentration levels greater than or equal to 0.89 mg/L, represents a 9 times higher risk of ED in MIS-C. Therefore, if confirmed by future studies, clinical assessment with newer promising biomarkers, such as Cys C, has the potential to determine which children warrant prioritization for cardiology referral or additional testing. Although it is difficult to reach definite conclusions, our data reveal that even clinically stable MIS-C patients face the potential of early vascular involvement. Evaluation of the association between the severity of MIS-C and outcome and Cys C levels would have provided important additional information. Our confirmation of Cys C as a CV risk marker does not shed further light on how it might influence pathobiological processes. Certainly, the interactions between Cys C and functional/structural microvascular changes in long-term MIS-C patients are complex and should be further explored, providing clinicians with a better grasp of the major mechanisms underlying the pathogenesis of ED. This may open up new possibilities to detect MIS-C patients with endothelial damage early to establish specific therapeutic interventions.

As mentioned above, in addition to the evidence of Cys C as an independent biomarker, other important findings were discovered. RHI values were significantly reduced (1.32 vs. 1.80; *p* = 0.001) and were related to Cys C of the MIS-C group. We confirmed a negative correlation of RHI with Cys C (r = −0.540; *p* = 0.006) and we found the statistically strongest cutoff at a Cys C value of 0.89 mg/L, with a specificity of 75%, a sensitivity of 75%, an odds ratio of 9, and 95% CL of 1.62–21.54. Finally, we confirmed a statistically significant Cys C factor, where higher Cys C values mean lower RHI values (*p* = 0.041). Furthermore, we observed that RHI also correlated with other biomarkers, but unlike the key predictive value of Cys C, they do not have such clinical significance, as no statistically significant difference was demonstrated between groups with RHI ≤ 1.4 and RHI > 1.4. These are a positive correlation of RHI with age (r = 0.508; *p* = 0.011), a negative correlation of RHI with ADMA (r = −0.428; *p* = 0.037), and a negative correlation of RHI with ALP (r = −0.555; *p* = 0.005). These data suggest that ADMA can also be a promising marker of ED in MIS-C patients.

The following correlations of Cys C were also confirmed, such as a negative correlation with ALT (r = −0.513; *p* = 0.010), a positive correlation with antiplatelet agents (r = 0.566; *p* = 0.004), and a positive correlation with SDMA (r = 0.458; *p* = 0.024), signifying that patients with higher SDMA tend to have higher Cys C values. A significant positive correlation emerged between creatinine and Cys C (r = 0.411; *p* = 0.046), marking that MIS-C patients with higher creatinine may tend to have higher Cys C values. These results may have clinical implications and might indicate that the risk of ED may be associated, in some parts, with changes in biomarker levels. However, there is no convincing epidemiological evidence to confirm these associations. We suggest that biomarkers may be contributing factors in the pathogenesis and maintenance of MIS-C. The role of biomarkers in the development of vascular changes merits further exploration. NT-proBNP, hsTnT, and CK are among clinical CV biomarkers, but recently sST2 has become a key prognostic marker of CVD, especially heart failure [32]. In our recent study, no correlation of sST2 with RHI or Cys C was demonstrated. Additional prospective studies should further address these possible associations, and the results of this study emphasize the importance of examining promising biomarkers in ED.

Growing evidence suggests that there exists a link between chronic inflammatory diseases activating systemic inflammatory biomarkers and increased incidence of CVD [33], suggesting that the chronic inflammatory process is a risk factor for vascular changes. In addition, our previous studies have shown the increased risk of ED due to a chronic inflammatory process [18,34], as well as supporting the claim that the RHI cutoff decreases with younger age [35].

MIS-C represents a hot topic at present in the pediatric population; however, the risk of long-term CV complications is still unknown. Although children are less severely affected, the severity of the clinical condition in MIS-C is mainly related to myocardial impairment and circulatory instability [5]. CV complications related to coronavirus disease 2019 (COVID-19) include, in addition to CV complications of hypercoagulable states, acute cardiac injury and inflammation [13]. Thrombotic complications are common with MIS-C, and patients with thrombotic complications have higher mortality [36]. Among the thrombotic complications, intracardiac thrombosis was demonstrated in one case in our cohort of MIS-C patients. Pei-Ni Jone et al. reviewed the current knowledge of MIS-C with a focus on CV manifestations and complications that are uncommon for children and young adults after COVID-19 disease or SARS-CoV-2 infection, according to a new scientific statement from the American Heart Association [37]. According to the literature, 50% of MIS-C patients have myocardial impairment, including depressed left ventricular function, coronary artery dilation or aneurysms, myocarditis, pericarditis, or refractory shock. In our study, transient heart involvement was confirmed in 17 patients (63%), including left ventricular dysfunction (37%), mitral and aortic valvulitis (14.8%), coronary artery dilation (14.8%), heart failure (7.4%), and arrhythmia (11.1%).

To date, in this context, ignoring potential confounding factors, we commend the authors of the pediatric study of microvascular dysfunction in MIS-C, which may, at least to some extent, be consistent with the findings of our study. Significant differences in RHI scores and a link between LVD and reduced ED were noted in the study of patients with MIS-C [38]. In our current study, RHI values were significantly lower in MIS-C (1.32 vs. 1.80; *p* = 0.001), and we found a statistically strongest clinical RHI cutoff in our cohort of 1.40, from which it follows that values ≤ 1.40 represent a 13.33 times higher risk of ED. With a cutoff of 1.67, RHI values ≤ 1.67 represent a 5.91 times higher risk (*p* = 0.005).

Although a long-term follow-up study is lacking and the prognosis of MIS-C requires further characterization, we can state that our overall impression from a longer-term point of view appears to be positive, as the majority of children experienced complete clinical recovery, with no significant medium- or long-term sequelae. The results of our study confirm that cardiac functions, measured by non-invasive methods, are within physiological limits in the medium-term interval from acute MIS-C.

### Strengths and Limitations

The main strength of the study was that we used RHI, which is an emerging non-invasive technique of early vascular changes, the homogeneity, and the full characterization of the study population, establishing well-defined selection criteria, the multivariable adjustment for several confounding factors, and a standard protocol for case ascertainment, thus eliminating the risk of selection bias. Another strength was that our inclusion criteria minimized the possible inclusion of children with only unresolved diagnoses not having MIS-C. Furthermore, it was the first study to demonstrate a potentially important role for Cys C in the clinical setting of MIS-C. This added to the comprehensiveness and reliability of the results. The study had potential limitations. This was a three-center study with a relatively small sample size, insufficient to analyze all relevant MIS-C outcomes, as is the case with most pediatric studies. Further studies with a larger cohort will be needed to clarify this issue. The disadvantage of this study lies in its observational nature, unable to prove causality. Furthermore, the likelihood of other residual or unnoticeable confounders could not be excluded, which may be a limitation to the applicability of the present findings. The generalizability of our findings to other ethnicities may be limited as our study population consisted primarily of Caucasian enrollees. Therefore, a large multiracial and multicenter study is required to confirm our results. Despite the limitations of the present study, just as our data found significant differences between the groups, we believe that these limitations did not compromise the current findings as to MIS-C. Thus, we assume our results concerning RHI in MIS-C are decent, fairly reliable, and representative, giving us confidence in extrapolating our findings for assessing microvascular changes in MIS-C. Taken together, our findings point out the clinical significance of further prospective research in children and adolescents with MIS-C to more accurately investigate ED and to answer the question of whether enhancement of ED should be considered a primary therapeutic endpoint.

## 5. Conclusions

This study established, for the first time, significant differences in ED in MIS-C subjects compared to HI without MIS-C with the cutoff value of RHI less than 1.40, suggesting reduced microvascular endothelial function. Our data suggested that Cys C levels were parallel with ED in MIS-C. Cys C level was an independent predictor of CVD risk in MIS-C patients. Thus, it is conceivable that the combination of these parameters may provide incremental prognostic significance, contributing to improved CVD risk stratification in MIS-C as well as to its prevention. Advanced impairment of endothelial function in peripheral microvasculature may be an important pathophysiological component of MIS-C. However, the immediate clinical applicability of RHI and Cys C as prognostic tools in patients with MIS-C requires further clinical and experimental studies. Early identification of ED can help reduce the incidence of CVD in patients with MIS-C and is clinically essential for a therapeutic approach to improve ED and clinical outcomes.

## Figures and Tables

**Figure 1 biomedicines-10-02956-f001:**
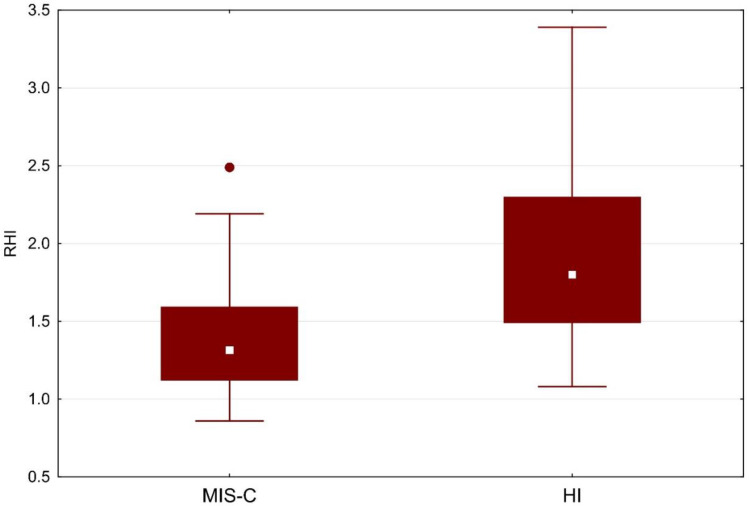
RHI in MIS-C vs. HI. Boxes indicate the interquartile range. White squares within boxes indicate medians. Whiskers extend to the highest or lowest values. HI: healthy individuals; MIS-C: multisystem inflammatory syndrome in children; RHI: reactive hyperemic index.

**Figure 2 biomedicines-10-02956-f002:**
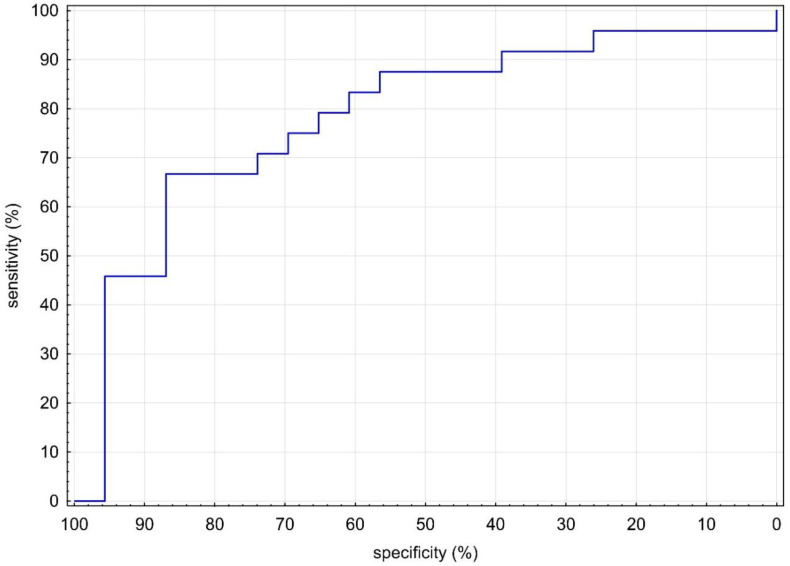
ROC curve of RHI levels between MIS-C and HI. HI: healthy individuals; MIS-C: multisystem inflammatory syndrome in children; RHI: reactive hyperemic index.

**Figure 3 biomedicines-10-02956-f003:**
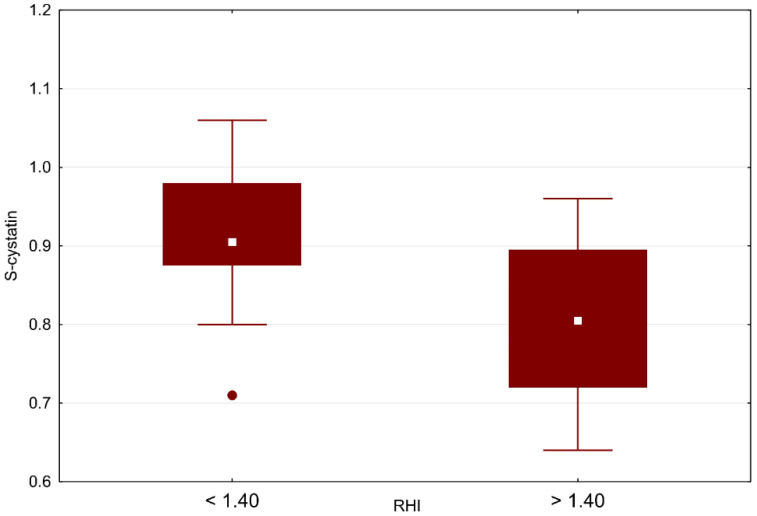
S-cystatin C vs. RHI cutoff for children with MIS-C. Boxes indicate the interquartile range. White squares within boxes indicate medians. Whiskers extend to the highest or lowest values. RHI: reactive hyperemic index.

**Figure 4 biomedicines-10-02956-f004:**
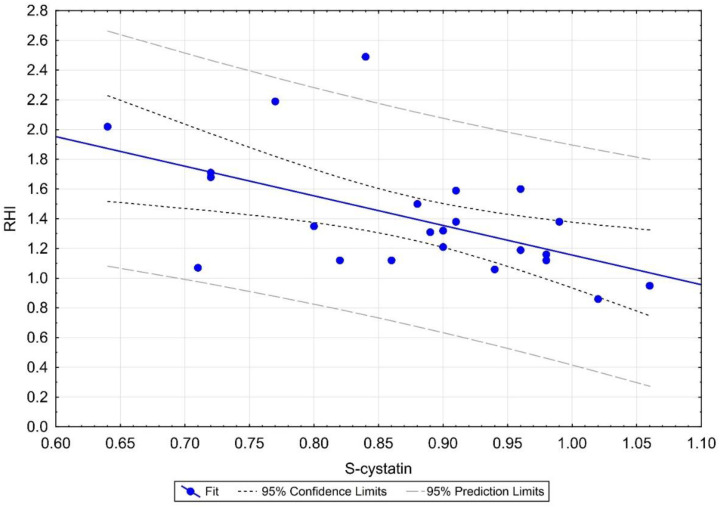
Linear regression of RHI and S-cystatin C in MIS-C. Moderately strong, negative correlation confirmed. MIS-C: multisystem inflammatory syndrome in children; RHI: reactive hyperemic index.

**Figure 5 biomedicines-10-02956-f005:**
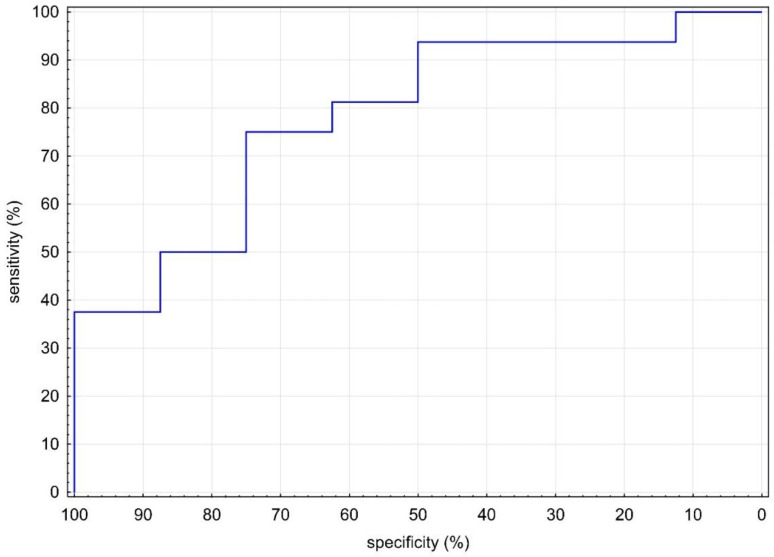
ROC curve of S-cystatin C levels in MIS-C group between RHI ≤ 1.4 and RHI > 1.4. MIS-C: multisystem inflammatory syndrome in children; RHI: reactive hyperemic index.

**Figure 6 biomedicines-10-02956-f006:**
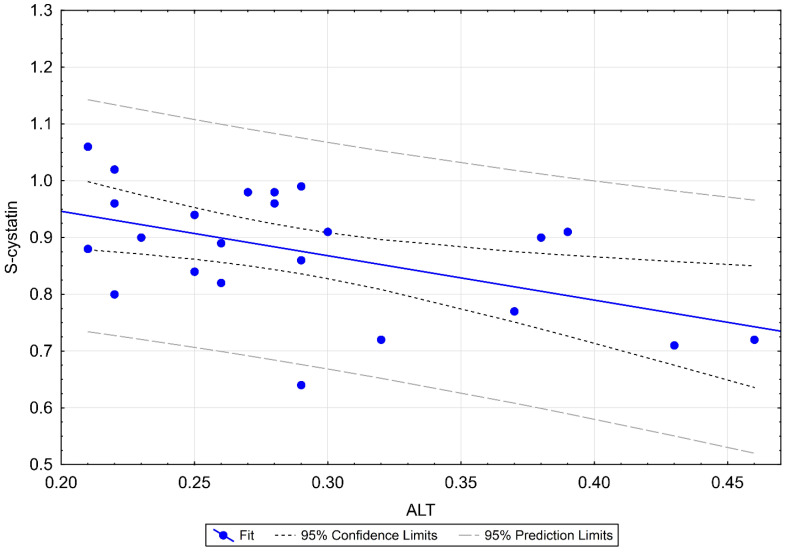
Linear regression of ALT and S-cystatin C in MIS-C. Moderately strong, negative correlation confirmed. ALT: alanine transaminase; MIS-C: multisystem inflammatory syndrome in children.

**Table 1 biomedicines-10-02956-t001:** Demographic data and RHI, and statistical analysis of MIS-C vs. HI.

Data	MIS-C Median (Min; Max)	Controls Median (Min; Max)	Statistical Significance
Number	27	23	
Gender (male/female)	15/9	13/10	Equivalency ±30%
Age (years)	12.76 (9.87; 20.01)	15.75 (12.36; 17.89)	Equivalency ±3
BMI (kg/m^2^)	20.82 (14.34; 29.36)	19.69 (16.23; 22.66)	Equivalency ±3
Systolic blood pressure (mmHg)	115 (90; 139)	116 (90; 129)	Equivalency ±10
Diastolic blood pressure (mmHg)	70 (60; 85)	66 (50; 75)	*p* = 0.009
RHI	1.32 (0.86; 2.49)	1.80 (1.08; 3.39)	*p* = 0.001

Values are expressed as the median with minimum and maximum values in parentheses. BMI: body mass index; HI: healthy individuals; MIS-C: multisystem inflammatory syndrome in children; NS: non-significant; RHI: reactive hyperemia index.

**Table 2 biomedicines-10-02956-t002:** MIS-C: file properties and correlations with RHI and Cys C.

Data	Median (Min; Max)	Statistical Significance	
		RHI	Serum Cys C
Age (years)	12.76 (9.87; 20.01)	r = 0.508; *p* = 0.011	NS (*p* = 0.728)
BMI (kg/m^2^)	20.82 (14.34; 29.36)	NS (*p* = 0.521)	NS (*p* = 0.075)
Systolic blood pressure (mmHg)	115 (90; 139)	NS (*p* = 0.380)	NS (*p* = 0.562)
Diastolic blood pressure (mmHg)	70 (60; 85)	NS (*p* = 0.112)	NS (*p* = 0. 850)
FEV_1_ (%)	99 (72; 127)	NS (*p* = 0.225)	NS (*p* = 0. 877)
Duration (month)	11.85 (3.62; 19.11)	NS (*p* = 0.083)	NS (*p* = 0.121)
ICU days (day)	2 (0; 11)	NS (*p* = 0.469)	NS (*p* = 0.2549)
Catecholamines (hour)	0 (0; 105)	NS (*p* = 0.391)	NS (*p* = 0.980)
Antiplatelets (day)	37 (0; 155)	NS (*p* = 0.101)	r = 0.566; *p* = 0.004
Anticoagulants (day)	8 (0; 15)	NS (*p* = 1.000)	NS (*p* = 0.280)

Values are expressed as the median with minimum and maximum values in parentheses. BMI: body mass index; FEV_1_: forced expiratory volume in 1 s; ICU: intensive care unit; NS: non-significant; RHI: reactive hyperemia index.

**Table 3 biomedicines-10-02956-t003:** Symptoms before and 3 months after MIS-C.

Symptoms	27 Patients		
	Yes	No	% Yes of Total
Physical performance—deterioration	10	17	37
Post-exercise difficulties (shortness of breath)	1	26	3.7
Fine motor skills—deterioration	0	27	0
Mental disorders	0	27	0
Sleep disorders	2	25	7.4
Sensory disturbances (smell, taste, hearing)	0	27	0
Joint ache	0	27	0
Others (palpitations, ankle swelling, nail peeling)	3	24	11.1

Columns Yes and No show the number of patients with a given symptom. Column % Yes of Total shows the percentage of patients with a given symptom.

**Table 4 biomedicines-10-02956-t004:** MIS-C: initial cardiac parameters.

	Data	Median (Min; Max)	Count	% of Total
Cardio markers	hsTnT (ng/L)	45 (3; 46,292)		
	NT-proBNP (ng/L)	2612 (35; 35,000)		
Echocardiography	Normal		13	48.1
	Heart involvement patients		14	51.9

Values are expressed as the median with minimum and maximum values in parentheses. hsTNT: high-sensitive troponin T; NT-proBNP: N-terminal fragment of brain-type natriuretic peptide; RHI: reactive hyperemic index.

**Table 5 biomedicines-10-02956-t005:** MIS-C: heart involvements.

Count	% of Total	Type of Involvements				
		Left Ventricular Dysfunction	Coronary Artery Dilation	Mitral and Aortic Valvulitis	Heart Failure	Arrhythmia
7	25.9	✓				
1	3.7		✓			
1	3.7		✓	✓	✓	
1	3.7	✓			✓	
2	7.4	✓	✓			
3	11.1			✓		
3	11.1					✓

The symbol “✓” indicates the presence of the type of heart involvement in the number of patients listed in the first column.

**Table 6 biomedicines-10-02956-t006:** MIS-C: biomarkers, cardiac, and laboratory parameters, and correlations with RHI and Cys C.

Parameters	Data	Median (Min; Max)	Statistical Significance	
			RHI	Serum Cys C
Cardiac	SF (%)	37.50 (27.00; 50.00)	NS (*p* = 0.456)	NS (*p* = 0.521)
	LVDd (%)	96.50 (70.00; 112.00)	NS (*p* = 0.619)	NS (*p* = 0.456)
	TAPSE (mm)	22.35 (9.00; 28.60)	NS (*p* = 0.093)	NS (*p* = 0.155)
	hsTnT (ng/L)	3.00 (3.00; 6.7)	NS (*p* = 0.983)	NS (*p* = 0.189)
	NT-proBNP (ng/L)	31.50 (10.00; 181.00)	NS (*p* = 0.511)	NS (*p* = 0.532)
Biomarkers	hsCRP (mg/L)	1.00 (1.00; 4.00)	NS (*p* = 0.710)	NS (*p* = 0.242)
	ADMA (μmol/L)	0.54 (0.32; 0.80)	r = –0.428; *p* = 0.037	NS (*p* = 0.092)
	SDMA (μmol/L)	0.62 (0.52; 0.76)	NS (*p* = 0.279)	r = 0.458; *p* = 0.024
	ADMA/SDMA	0.93 (0.42; 1.19)	NS (*p* = 0.060)	NS (*p* = 0.344)
Laboratory parameters	Ferritin (µg/L)	30.30 (7.40; 178.1)	NS (*p* = 0.476)	NS (*p* = 0.266)
	D-dimer (mg/L FEU)	0.24 (0.22; 0.76)	NS (*p* = 0.957)	NS (*p* = 0.193)
	Fibrinogen (g/L)	2.49 (1.62; 3.71)	NS (*p* = 0.969)	NS (*p* = 0.731)
	GDF-15 (ng/L)	426.15 (400.00; 1241.00)	NS (*p* = 0.752)	NS (*p* = 0.326)
	IL-6 (ng/L)	N/A *	N/A *	N/A *
	Procalcitonin (µg/L)	0.02 (0.02; 0.05)	NS (*p* = 0.500)	NS (*p* = 0.727)
	sST2 (μg/L)	14.40 (5.76; 32.12)	NS (*p* = 0.545)	NS (*p* = 0.509)
	Na (mmol/L)	140 (137; 144)	NS (*p* = 0.671)	NS (*p* = 0.161)
	K (mmol/L)	4.35 (3.90; 4.80)	NS (*p* = 0.574)	NS (*p* = 0.413)
	Cl (mmol/L)	105 (101; 107)	NS (*p* = 0.577)	NS (*p* = 0.140)
	S-creatinine (µmol/L)	62 (43; 103)	NS (*p* = 0.611)	r = 0.411; *p* = 0.046
	S-cystatin C (mg/L)	0.90 (0.64; 1.06)	r = –0.540; *p* = 0.006	N/A
	Bilirubin (µmol/L)	7.50 (3.00; 24.00)	NS (*p* = 0.167)	NS (*p* = 0.378)
	ALT (µkat/L)	0.28 (0.21; 0.46)	NS (*p* = 0.220)	r = –0.513; *p* = 0.010
	ALP (µkat/L)	3.77 (0.68; 5.51)	r = –0.555; *p* = 0.005	NS (*p* = 0.206)
	S-albumin (g/L)	48.95 (45.10; 53.80)	NS (*p* = 0.736)	NS (*p* = 0.424)
	CK (µkat/L)	1.86 (1.05; 4.36)	NS (*p* = 0.314)	NS (*p* = 0.604)
	LD (µkat/L)	3.70 (2.55; 5.87)	NS (*p* = 0.418)	NS (*p* = 0.599)
	U-Na (mmol/L)	149.50 (31.00; 291.00)	NS (*p* = 0.758)	NS (*p* = 0.197)
	U-K (mmol/L)	59.50 (10.00; 186.00)	NS (*p* = 0.186)	NS (*p* = 0.333)
	U-Cl (mmol/L)	182 (52; 310)	NS (*p* = 0.450)	NS (*p* = 0.705)
	U-creatinine (mmol/L)	12.56 (2.79; 37.61)	NS (*p* = 0.700)	NS (*p* = 0.749)
	U-albumin (mg/L)	7 (3; 448)	NS (*p* = 0.473)	NS (*p* = 0.651)
	U-ACR (g/mol)	0.75 (0.40; 11.90)	NS (*p* = 0.541)	NS (*p* = 0.741)
	Leukocytes (10^9^/L)	6.33 (3.60; 10.83)	NS (*p* = 0.822)	NS (*p* = 0.736)
	Lymphocytes abs (10^9^/L)	2.25 (1.20; 4.10)	NS (*p* = 0.303)	NS (*p* = 0.572)
	Neutrophiles abs (10^9^/L)	2.95 (1.60; 6.70)	NS (*p* = 0.730)	NS (*p* = 0.954)

Values are expressed as the median with minimum and maximum values in parentheses. ADMA: asymmetric dimethyl arginine; ALP alkaline phosphatase; ALT: alanine transaminase; BMI: body mass index; CK: creatine kinase; FEU: fibrinogen-equivalent units; GDF-15: growth differentiation factor-15; hsCRP: high-sensitive C-reactive protein; hsTNT: high-sensitive troponin T; IL-6: interleukin-6; LD: lactate dehydrogenase; LVDd: left ventricular diastolic diameter; N/A: not applicable; NS: non-significant; NT-proBNP: N-terminal fragment of brain-type natriuretic peptide; SDMA: symmetric dimethyl arginine; SF: shortening fraction; sST2: suppression of tumorigenicity 2; TAPSE: tricuspid annular plane systolic excursion; U-ACR: urine albumin-creatinine ratio. * All examined MIS-C patients had IL-6 values under the limit of quantification of the method, i.e., <1.5 ng/L.

## Data Availability

The data presented in this study are available upon request from the first author for privacy reasons.

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
