# Peer review of "Circulating Serum Cystatin C as an Independent Risk Biomarker for Vascular Endothelial Dysfunction in Patients with COVID-19-Associated Multisystem Inflammatory Syndrome in Children (MIS-C): A Prospective Observational Study"

_biomedicines, 2022, doi:10.3390/biomedicines10112956_

Round 1

Reviewer 1 Report

The results of study seemed to be interesting, and the paper was well prepared.

1.      In this study, the inflammatory markers such as CRP and leukocytes were not clearly related to Cys C. Is Cys C a potential inflammatory marker as stated by the authors? Please speculate more clearly about the phenomenon of Cys C to systemic inflammatory markers in the study.

2.      How were the healthy controls recruited? In general, there are some difficulties in ethical views to recruit such controls in cases of non-diseased children. Please describe in more details about the recruitment and criteria.

3.      How was the family history (described in Methods) collected and determined? Please describe more about the accuracy of information.

4.      In general, pressure in arm and rest during the test are barriers to apply the Endo-PAT test to children. Did the study give something to make the test working in children? Was the test performed completely to ‘all’ children?

5.      The coefficient of variation of important and research-specific tests such as Cys C, ADMA, SDMA, and cytokines etc. should be added to the revision.

6.      Adverbs such as however and therefore were used like conjunctions in some parts. Please correct them.

Author Response

Revisions to Manuscript No.: 2036848

Dear Reviewer,

Cordial greetings and many thanks for your thorough revision of our manuscript and valuable remarks. We appreciate your helpful suggestions and positive criticism. We took them into consideration and revised the text. The manuscript was reviewed by a native speaker and updated accordingly.

All point-by-point changes are listed below with an explanation of the modifications and are marked up using  the “Track Changes“ feature.

We are looking forward to further collaboration in the future.

With all best wishes,

Marcela Kreslova

REVIEWER 1

Comments and Suggestions for Authors

The results of study seemed to be interesting, and the paper was well prepared.

Authors’ Responses

We appreciate the reviewer’s helpful suggestions, comments and positive criticism. We would like to thank the reviewer very much for general comments on our manuscript. We found them very useful and interesting. We took them into consideration and updated the manuscript accordingly.

The following is a point-by-point response to the reviewer´s comments and concerns:

1) In this study, the inflammatory markers such as CRP and leukocytes were not clearly related to Cys C. Is Cys C a potential inflammatory marker as stated by the authors? Please speculate more clearly about the phenomenon of Cys C to systemic inflammatory markers in the study.

We thank the reviewer and we fully agree with this valuable remark.

We modified  a part of chapter 4. Discussion. We suggest that it could be potentially plausible that the unifying concept between Cys C as a potential inflammatory marker [28] and CVD might be micro-inflammation [9,29,30]; however, the role of Cys C and the possible connection of inflammatory pathways needs to be further clarified before definitive conclusions can be made.

We also added a new reference by Demirkol et al. which proposes that Cys C levels should correlate with ED and inflammation in-directly through renal function [27].

2) How were the healthy controls recruited? In general, there are some difficulties in ethical views to recruit such controls in cases of non-diseased children. Please describe in more details about the recruitment and criteria.

We thank the reviewer for a very useful comment. 

We recruited children who came to the pulmonology clinic and in whom no chronic respiratory disease was detected. They had problems of a non-somatic nature or they had only mild nasal allergic symptoms. It was confirmed by a questionnaire system in the presence of a physician that these children were not being treated for any inflammatory, neoplastic, metabolic, cardiac or peripheral vascular disease, as well as not taking any anti-inflammatory, antibiotic or vasoactive treatment.
These children and their parents were offered to participate in the study and the course of the examination was explained to them in detail. Children were measured after signing informed consent.

3) How was the family history (described in Methods) collected and determined? Please describe more about the accuracy of information.

We thank the reviewer for the valuable remark.

We used a detailed questionnaire system with the help of an experienced cardiologist to collect the necessary data on family CVD risk.

Patients and their parents filled out a detailed questionnaire and answered questions about family and personal history. We included healthy subjects with no history of inflammatory, metabolic, or neoplastic diseases. None of them reported a history of heart disease, treatment with antibiotics, anti-inflammatory drugs, or drugs with known adverse effects on endothelial function. Exclusion criteria were also dyslipidemia, arterial hypertension, positive family history of premature cardiovascular events, abnormal left ventricular function, and smoking.

A positive family history of premature manifestation of CVD was defined as the occurrence of sudden death or myocardial infarction in the father or a first-degree male relative before the age of 45, and the occurrence of sudden death or myocardial infarction in the mother or a first-degree female relative before the age of 55.

The relevant information was added to chapter 2.1. Study population.

4)  In general, pressure in arm and rest during the test are barriers to apply the Endo-PAT test to children. Did the study give something to make the test working in children? Was the test performed completely to ‘all’ children?

Thanks to the reviewer for these questions.

The course of the examination was explained in detail to all patients and their parents, with an appeal to temporary discomfort from the inflated cuff. An empathetic approach of the investigator was necessary and the presence of the parent was an advantage (especially for younger children). The children were also motivated by a small reward at the end of the trial.

In our study, the examination could not be successfully completed in three patients out of a total of 27 children - for technical reasons (leaky sensors in one small child) and negative subjective feelings of two patients during inflation of the pressure cuff on the arm.

The number of overall examined children and successfully completed measurements was already mentioned in the manuscript in chapter 3.2. Cardiac outcomes.

5) The coefficient of variation of important and research-specific tests such as Cys C, ADMA, SDMA, and cytokines etc. should be added to the revision.

We thank the reviewer for the valuable remark and we fully agree with it. We added the coefficients of variation into the manuscript in chapters 2.5.1. ADMA and SDMA quantification, 2.5.2. Special parameters determination and 2.5.3. Routine parameters determination.

6) Adverbs such as however and therefore were used like conjunctions in some parts. Please correct them.

We thank the reviewer for this remark. We checked the spelling, adjusted the conjunctions and adverbs. The manuscript was also reviewed by a native speaker.

We would like to once again thank the reviewer for invaluable help in the revision of our manuscript and for the remarks.

Reviewer 2 Report

Extremely interesting subject, the bibliography shows the authors' interest in the subject, the design of the study clearly presented, concise, well structured.

Clear discussions, pertinent conclusions./

I have inserted some of my observations directly into the text

Author Response

Dear Reviewer,

Cordial greetings and many thanks for your thorough revision of our manuscript and valuable remarks. We greatly appreciate your helpful suggestions which we have taken into account and incorporated into the text. Thank you very much and we appreciate your very positive feedback.

We have taken the necessary measures to revise the text again. The manuscript was reviewed by a native speaker and updated accordingly.

All changes are marked up using the “Track Changes“ feature.

We are looking forward to further collaboration in the future.

With all best wishes,

Marcela Kreslova

REVIEWER 2

Comments and Suggestions for Authors

Extremely interesting subject, the bibliography shows the authors' interest in the subject, the design of the study clearly presented, concise, well structured.

Clear discussions, pertinent conclusions.

I have inserted some of my observations directly into the text

Authors’ Responses

We are very grateful to the reviewer for his/her very positive assessment and appreciation of the quality of the structured discussion and conclusion. We appreciate the reviewer's useful suggestions and observations, especially inserting specific notes directly into the text. We took them into account and updated the manuscript.

The manuscript was reviewed by a native speaker and updated accordingly.

All changes and modifications are marked in the text using the "Track changes" function.

Commented [M3]: more details....which catecolamines?
Commented [M5]: which catecolamines?

The types of catecholamines and anticoagulants with percentages are described in detail in chapter 3.1. Demographic data of the study population.

We would like to thank the reviewer once again for his/her invaluable help in the revision of our manuscript.

Round 2

Reviewer 1 Report

The report was much improved.